# A Hierarchical Age–Period–Cohort Analysis of Breast Cancer Mortality and Disability Adjusted Life Years (1990–2015) Attributable to Modified Risk Factors among Chinese Women

**DOI:** 10.3390/ijerph17041367

**Published:** 2020-02-20

**Authors:** Sumaira Mubarik, Fang Wang, Saima Shakil Malik, Fang Shi, Yafeng Wang, Chuanhua Yu

**Affiliations:** 1Department of Epidemiology and Biostatistics, School of Health Sciences, Wuhan University, Wuhan 430071, China; sumairaawan86@gmail.com (S.M.); wangfang0923@whu.edu.cn (F.W.); 18204313963@163.com (F.S.); wonyhfon@whu.edu.cn (Y.W.); 2Department of Zoology, University of Gujrat, Gujrat 50700, Pakistan; saimamalik25@yahoo.com; 3Department of Nutrition and Food Hygiene, School of Health Sciences, Wuhan University, Wuhan 430071, China; nawshermkd177@gmail.com; 4Global Health Institute, Wuhan University, Wuhan 430071, China

**Keywords:** mortality rates, age, cohort, breast cancer, high body mass index, China

## Abstract

Limited studies quantified the age, period, and cohort effects attributable to different risk factors on mortality rates (MRs) and disability-adjusted life years (DALYs) due to breast cancer among Chinese women. We used data from the Global Burden of Disease Study (GBD) in 2017. Mixed-effect and hierarchical age–period–cohort (HAPC) models were used to assess explicit and implicit fluctuations in MRs and DALYs attributable to different breast cancer associated risk factors. As the only risk factor, high body mass index (HBMI) showed continuously increasing trends in MRs and DALYs across ages, periods, and cohorts. Age, recent periods (2010–2015), and risk factor HBMI showed significant positive effect on MRs and DALYs (*p* < 0.05). Moreover, we reported significant interaction effects of older age and period in recent years in addition to the interplay of older age and risk factor HBMI on MRs and DALYs. Increased age and obesity contribute to substantially raised breast cancer MRs and DALYs in China and around the globe. These discoveries shed light on protective health policies and provision of healthy lifestyle for improving the subsequent breast cancer morbidity and mortality for China, as well as other related Asian regions that are presently facing the same public health challenges.

## 1. Introduction

Breast cancer is the most commonly diagnosed cancer or disease among women all over the world and linked with considerable years of life lost, leading to increased cancer-related mortality and morbidity [1,2,3]. Breast cancer has multifactorial etiology and involves a complex interplay between genetic, epigenetic, and adjustable lifestyle or environmental factors [4]. Considering lifestyle factors, obesity, extravagant alcohol consumption, physical inactivity, and insufficient diet protrude as adaptable risk factors, which, if avoided, could help in breast cancer management and prevention [5]. It is still unknown whether researchers explored the burden of mortality rates from breast cancer due to all these adaptable risk factors among the individuals of the same population.

Age is one of the most imperative breast cancer incidence-related risk factors. The breast cancer incidence varies drastically with race and ethnicity, even among young women. White women are faced increasing breast cancer risk in age above 45 years in comparison with black women. Older breast cancer patients face poor survival rates than young ones, and studies demonstrated that older age was an independent prognostic factor for adverse disease outcomes [6,7].

Obesity affects 13% of the population in the world or more than six hundred million adults around the globe, and it is defined as a body mass index (BMI) ≥30 kg/m^2^ [8]. Obesity is a major health issue with devastating impacts in various medical conditions like cancer, diabetes, and cardiac issues [9,10,11]. A wide range of observational studies showed that almost 96% of breast cancer cases increase weight during the course of treatment, with the finding that post-diagnosis weight gain has an inverse relationship with disease-free survival [5,12,13,14]. Higher weight gain is reported among chemotherapy-treated premenopausal cases along with those who are obese or overweight at time of diagnosis [15]. Obesity is linked with shorter time to disease recurrence and increased mortality for both pre- and postmenopausal breast cancer cases. One of the cancer prevention studies by the American Cancer Society illustrated statistically significant association between obesity (higher BMI) and breast cancer mortality after following almost 0.5 million women for a period of 17 years (1982–1998). Moreover, they reported that patterns of overweight and obesity in the United States could account for 20% of all deaths from cancer in women [16]. The Contraceptive and Reproductive Experiences study reported the association of obesity with increased breast cancer-specific mortality among white women, but results were non-significant in African American women [17,18].

A meta-analysis demonstrated a strong association between physical inactivity and increased risk of breast cancer [19]. However, some contradictory results were also reported in the literature. Loprinzi et al. reviewed 76 studies on the association of physical activity with breast cancer risk, and, among them, 47% of studies were unable to find any protective effect [20]. The Cancer Prevention Study-II Nutrition Cohort recently illustrated no association of pre- and post-diagnosis physical activity with breast cancer-specific mortality [21].

None of the studies reported the association of any single specific factor with breast cancer while evaluating the global burden of disease morbidity and mortality, and they concluded that multiple factors are responsible for the onset and development of breast cancer [3,22]. Epidemiological monitoring and evaluation of disease morbidity and mortality can only be done after studying all the possible disease-related factors or causes. However, keeping a record of risk factors does not quantify the exact disease burden [3,4].

Although breast cancer incidence is lower in China as compared to Western countries, China is currently facing increasing trends in its incidence and mortality [23]. In view of inadequate medical resources, specifically in rural areas of China, a risk prediction model that is appropriate for general population screening is immediately required. Therefore, the current study aimed to estimate breast cancer mortality and disability-adjusted life years (DALYs) across age, period, and cohort based on different risks including higher BMI, low physical activity, smoking, and alcohol consumption. In addition, the objective of the current study was to measure explicit and implicit fluctuations in breast cancer mortality rates (MRs) and DALYs due to the interaction of different risk factors in Chinese women by using a hierarchical age–period–cohort (HAPC) model.

## 2. Materials and Methods

### 2.1. Data Source

The Global Burden of Disease (GBD) study in 2017 included a yearly evaluation casing 195 countries and territories from 1990 to 2017. It covered 476 risk–outcome cases including 359 diseases and injuries, with 84 behavioral, environmental, occupational, and metabolic risks or groups of risks from 1990 to 2017 by age, sex, year, and location. It extracted relative risk and exposure estimates from 46,749 randomized controlled trials, cohort studies, household surveys, census data, satellite data, and other sources. There are two primary sources of data for China: surveillance data from the China Disease Surveillance Points (DSP) system and vital registration (VR) data collected by the Chinese Center for Disease Control and Prevention (CDC). Gagkidu et al. explained the methodology in detail along with the approach of the GBD 2017 [24].

We only extracted the data of the Chinese female population from this study. From 1990 to 2015, the data for association between breast cancer MRs and DALYs with various risk factors like age (20–80) and period (1990–2015), as well as classified risk factor variable with classes of high body mass index (BMI >30 kg/m^2^ for adults), alcohol use, low physical activity (Low PA), and smoking, were extracted for women aged 20–84 years old.

### 2.2. Outcomes

The outcomes of interest were MRs and DALYs from breast cancer patients (measured as per 100,000 population). Mortality data source of breast cancer from vital registration (VR), cancer registration systems, and verbal autopsy (VA) data, and mortality estimates were used as input data into the Cause of Death Ensemble Model (CODEm). The CODEm predicted MRs based on all available data, and then the COD correct algorithm was used for each single cause estimation.

DALYs are a summary measure of population health widely used to quantify disease burden; they are composed of years lived with disability (YLDs) and years of life lost due to premature mortality (YLLs). YLLs represent the mortality component of DALYs, and YLLs due to breast cancer were calculated by using standard life expectancy and the number of deaths according to age. YLDs, the morbidity component of the DALYs, were calculated by multiplying the prevalence (number of cases) of each sequela by its disability weight. For a given risk–outcome pair, the attributable DALY estimate was obtained by multiplying the total DALYs by the population attributable fraction (PAF) for the risk–outcome pair for each age, sex, location, and year.

As the aim of the study was quantifying the variations in breast cancer MRs and DALYs across age, periods, cohorts, and interactions with different risk factors, an individual model was constructed for both outcomes for the Chinese female population.

### 2.3. Predictors

The GBD study was based on adjustable breast cancer risk factors, i.e., the justification of data quality and the eminence of the estimate models [24]. MR and DALY values were retrieved for women aged 20–84 years, between 1990 and 2015, with a five-year interval. The statistical analysis period was categorized as a categorical variable and year 1990 was considered as the benchmark or reference category. Furthermore, a classified risk factor variable was included, which consisted of alcohol use, high body mass index (HBMI), low physical activity (Low PA), and smoking, defined as dummy variables with alcohol use as a reference category. A detailed description of the modeling and definition used for the risk factors was previously published [25]. Furthermore, China’s surveys for all these risk factors are also available at http://ghdx.healthdata.org/gbd-results-tool.

### 2.4. Statistical Analysis

To measure the potential changes related to risk factors, we described the breast cancer outcomes (MRs and DALYs) stratified by different risks for age, period, and cohort. Furthermore, in order to assess the variation and median trend of mortality and DALYs irrespective of risk factors (for all risk factors together), we also reported the boxplot by age, period, and cohort for each outcome.

A sequence of explicit assessments of variances in MRs and DALYs within individuals across different ages (age effect) and population-wide changes in MRs and DALYs over time (period effect) were measured by successions of longitudinal mixed-effect models (MEMs) with random and fixed effects of individual-level and random coefficients. Furthermore, implicit assessments of the random variances (effects) in outcomes connected to different risks across ages, periods, and cohorts were also performed. We selected the MEMs because of their ability to handle asymmetric data and non-discrete covariates. Moreover, the identifiability problem could be overcome by the hierarchical structure of the models with no assumption of linear and additive effects of age–period–cohort (APC) at the same level of analysis [26,27,28].

MEMs also permit individuals (observations) to have their coefficients on different levels, whereby some observations are above and some below the population mean (regression intercept); similarly, some individuals have a higher rate of change than the population mean (regression slope) and some have a lower rate of change than population mean. In this situation, the model accommodates more flexibility and can estimate the differences in outcome at different levels and over time in the presence of heterogeneity. In the present study, three models were employed. Among them, model 1 contained age, levels of period (year), and dummy-classified risk factor variables with four risks. The years and risk factors coefficients described the period and risks effect, respectively. Model 2 included the additional age interaction with year in model 1. Interaction terms allowed us to assess how effects of period may fluctuate with age (cohort effect). Model 3 involved the interaction of age and risk factors by extending model 2. We aimed to evaluate in what way these interactions described some of the period effects, plus how risk factor effects may vary through age. In order to provide information about how these variables were linked to MRs and DALYs, we compared these models with the traditional generalized linear model (GLM) by including the full set of variables. The goodness of fit was evaluated using the Akaike information criterion (AIC), Bayesian information criterion (BIC), and log-likelihood statistic for the estimated models.

A hierarchical age–period–cohort (HAPC) analysis of longitudinal panel data [26,29,30], explicitly incorporating age, period, and cohort random effects, and indirectly incorporating these effects via interaction with risks to these factors, was used to confirm the results obtained by MEMs. All analysis was conducted using R package version 3.5.2 [31]. Results were considered significant with *p* < 0.05.

## 3. Results

Longitudinal analyses of mortality rates (MRs) and DALYs due to breast cancer across ages attributable to different risk factors like high body mass index (HBMI), alcohol use, low physical activity (Low PA), and smoking for different years among Chinese women are depicted in Figure 1.

Rising trends of MRs and DALYs were noticed with increased age in all risk factors under consideration. In particular, higher MRs and DALYs were observed for the risk factor HBMI from 1990 to 2015; specifically, the age groups of 50 to 75 years for MRs and 50 to 60 years for DALYs were more stressed from 2000 onward. Mortality and DALYs attributable to HBMI peaked in 2015 as compared to previous years in the age groups of 70–75 years and 55–60 years, respectively. These rates were almost three times greater than the MRs of 2010 to 2015. On the other hand, at young ages, low breast cancer MRs and DALYs were observed with reference to HBMI (Figure 1). The median trend of MRs and DALYs irrespective of risk factors (for all risk factors together) by age, period, and cohort are given in Appendix A. Together, this evidence supports the existence of confounding by risk (HBMI) on secular increases in MRs and DALYs in Chinese women and justifies subsequent HAPC analyses.

### 3.1. Mortality Rates (MRs)

The age-, period-, and cohort-related nonlinear patterns in MRs with respect to various risk factors were confirmed using the longitudinal analysis (Figure 1 and Appendix A). An age-square term was not considered in the models because, when we included age-square term, the coefficients for both variables were contrasted with each other. Moreover, inclusion of the age-square term renders the actual age effect insignificant with a negative coefficient, but considering only actual age gives a significant positive association of age and MRs by assuming the impact of other variables as constant. When adjusting for age, it was noted that the period effect was positively related to MRs in the year of 2015. The findings revealed that the HBMI effect was positively related with MRs when adjusting for age and period. Moreover, a significant interaction effect of age and period was observed in 2010 and 2015 with MRs (*p* < 0.01), as well as a significant interaction effect of age and HBMI with MRs (*p* < 0.01), when controlling for other factors (Table 1, model). In model 1, age and HBMI were found as positive predictors of MRs. A significant period effect on MRs was observed in 2005, 2010, and 2015 (Table 1, model 1). In model 2, we found that the addition of an interaction of age and period resulted in an almost doubled change in MRs by 0.010, 0.020, and 0.029 in 2005, 2010, and 2015 respectively. Upon subsequently adjusting for potential confounding effects of age and period in 2010 and 2015, HBMI, and the interaction of age and period in 2005, 2010, and 2015, the interaction of age and HBMI was identified as a positively associated factor with MRs (Table 1, model 3). In order to compare the coefficients among models, we demonstrated that the effect of HBMI was significantly greater, and age accounted for most of the period effects and HBMI. Moreover, model selection parameters were estimated and based on low AIC and BIC values. Model 3 was considered a good fit for MRs from breast cancer (Table 1).

### 3.2. Disability-Adjusted Life Years (DALYs)

Longitudinal analysis of DALYs due to breast cancer across age, period, and cohort based on different risks depicted a non-linear pattern of age, period, and cohort in DALYs (Figure 1 and Appendix A). The age-, period-, and risk-related variations in DALYs were identified by mixed-effect models. Advanced age, recent periods, and HBMI were positively associated with DALYs. Moreover, the interaction effects of age and period in 2015 and age and HBMI were significantly associated (*p* < 0.05) with DALYs (Table 2, model 1). Results from model 1 illustrated a positive association of age and period with DALYs in three different years including 2005, 2010, and 2015.

Moreover, HBMI showed a significantly positive association (31.7, *p* < 0.001) with DALYs as compared to the reference group (Table 2, model 1). The addition of an interaction effect of age and period in model 2 provided a significant positive interaction effect of age and period with DALYs in 2010 and 2015 (Table 2, model 2).

On the other hand, inclusion of the interaction between age and risks showed a significantly positive effect of the interaction of age and HBMI with DALYs. Controlling for the other effects, a substantial positive HBMI effect on DALYs was observed that was approximately seven times higher than the reference coefficient (Table 2, model 3). To compare the coefficients among models, we again confirmed that the effect of HBMI was significantly greater and that the interaction of age and period in recent years, in addition to the interaction of age and HBMI, was higher relative to other effects. Furthermore, model selection parameters were estimated and based on low AIC and BIC values. Model 3 was considered a good fit for DALYs from breast cancer (Table 2).

### 3.3. Random Effects of Age, Period, Cohort, and Interaction with Risk Factors

Random effects of age, period, cohort, and interaction with risk factors by the selected HAPC model provide evidence of how different ages, years, and cohorts cause an increase in MRs and DALYs and interact with the various risk factors. We observed an increasing trend of coefficient in the random effect of age from 50 to 80 years, which suggested that older ages were positively associated with MRs from breast cancer, and a higher random effect of years was observed in 2010 and 2015. The cohort 1910 to 1935 showed a rising trend of random cohort effects in MRs; additionally, a decreasing trend of random effects was reported in the overall cohort (1910 to 1970), suggesting that the early cohorts were positively associated with MRs due to breast cancer (see Appendix A). Moreover, a considerably higher random effect of age was found from 55 to 75 years of age. A substantial association was observed between older ages and DALYs, particularly for women with ages ranging from 55 to 65 years old. Similarly, a rising trend of period random effects in DALYs was observed with a higher increase in 2010 to 2015, and the overall cohort 1910–1970 showed a declining trend of random cohort effects, signifying a positive association of cohort 1910–1940 with DALYs from breast cancer (Appendix A).

Furthermore, the random effects of individual age, period, and cohort in each risk were reported to assess the interaction of different levels of age, period, and cohort with various risks. Results indicated that women with older ages from 50 to 75 years had HBMI, which was found to be a significant risk factor of MRs. Similarly, extensive higher random effects of period and risks on MRs were identified in a period of 10 years from 2005 to 2015 among women who had HBMI. The interaction of cohorts and risks showed a declining trend of random effects on MRs from the overall cohort 1910–1965 among women with HBMI and rising trends in early cohorts, i.e., 1910–1935 (Figure 2). On the other hand, considerably higher random effects of the interaction of age and HBMI on DALYs were observed in women with ages from 55 to 65 years, while a continuously increasing trend of random effects of interaction between period and HBMI was noted throughout the year, with a particularly greater effect on DALYs detected in the year 2005–2015. On the other hand, higher random effects of interaction of cohort and HBMI were perceived on DALYs in the cohort 1910–1940 with an overall declining trend of cohort and HBMI interaction in the cohort 1910–1965 (Figure 3).

In associating the random effects among age, period, cohort, and interaction with risks, we again confirmed that all random effects were higher in women with HBMI, and older ages, recent years, and early cohorts contributed to increasing MRs and DALYs attributable to HBMI.

### 3.4. Fixed and Random Effects Comparison

The comparison of fixed and random effects across ages, periods, and cohorts is displayed in Figure 4 for MRs and DALYs. The two sets of coefficients, i.e., fixed and random effects, yielded almost similar estimates of trends across the age, period, and cohorts. In addition, random effects also depicted that there were reduced unexplained sources of variation in breast cancer MRs and DALYs as compared to explained sources of variation or fixed effects. Therefore, it can be assumed that the random effects are independent of regressors, and that fixed and random effects were parallel for the MR and DALY dataset (Figure 4).

## 4. Discussion

It is well documented that, over the past two decades, the prevalence of obesity tripled (11%–29%) among Chinese adults [32]. In the present study, longitudinal analysis revealed that MRs and DALYs among HBMI breast cancer cases doubled in the age group of 50 to 75 years and 50 to 65 years for deaths and DALYs, respectively. Correspondingly, continuously increasing trends of deaths and DALYs due to breast cancer were observed from 1990 to 2015 in women with HBMI, and they tripled from 2010 to 2015. Researchers reported different results in various geographical regions depending upon ethnicity and genetic differences. An increased breast cancer mortality rate was estimated among Europeans by 2020. Even though breast cancer prevalence is higher in developed countries, greater MRs were observed in underdeveloped regions [33], possibly due to limited access to proper diagnosis and treatment. Moreover, almost 90% of deaths were reported in the United States due to breast cancer among women with 50 years of age or older [34]. Therefore, increased age and HBMI are considered as potential breast cancer risk factors [33] and linked with higher death rates and DALYs.

In the current study, controlling for age, the period effect was found to be positively associated with MRs in 2015. Additionally, results of the model presented that HBMI was positively associated with MRs when adjusting for age and period, and a significant interaction effect of age and period was observed in 2010 and 2015 with MRs, along with a significant interaction effect of age and HBMI with MRs when controlling for other factors. One of the Indian epidemiological studies reported increased age-standardized breast cancer incidence rate (39%) over 26 years (1990–2016), and this increase was observed in every single state. HBMI, high plasma glucose levels, and passive smoking were considered to have roles in breast cancer DALYs [35]. Youlden et al. demonstrated increased age-specific breast cancer incidence rates among Australian women aged 50 to 69 years, overlapping with the target age range for disease screening. Breast cancer incidence rates also increased with increased age in the Philippines [36].

The current study demonstrated that, in associating the random effects among age, period, cohort, and interaction with risks, it was established that all random effects were higher in women with HBMI, and that older ages, recent years, and early cohort contributed to increasing MRs and DALYs in relation to HBMI. Moreover, both random- and fixed-effect models confirmed similar results to strengthen the findings.

Biological mechanisms underlying the relationship between breast cancer and obesity clarify the reasons that obesity contributes to DALYs and deaths. Excess circulating hormones are present, particularly estrogen, which causes the development of carcinogenesis and mutations by prompting free-radical production, revealing genotoxicity. Hyperinsulinemia is another factor causing breast cancer, which encourages insulin-like growth factor (IGF–1) production and activity [37]. Therefore, increased accumulation of estrogen and IGF–1 contributes to breast cancer development and progression. Studies reported a strong association of obesity with shorter time to disease recurrence and increased mortality for both pre- and post-menopausal breast carcinoma. One of the multicenter studies reported a significant association of obesity with increased breast cancer-specific mortality, regardless of ethnicity, among American women older than 50 years of age [38]. None of the studies that assessed the MRs and DALYs due to breast cancer attributable to different risk factors, especially those related to obesity, were found in China. The current study enhances the increasing body of evidence that healthy years of life are lost due to breast carcinogenesis, not only from all causes but precisely because of obesity.

The current study has certain limitations such as the unavailability of data for all the risk factors including alcohol intake, physical inactivity, use of oral contraceptives, and socioeconomic status. Moreover, if exact BMI and physical inactivity values (proper measures) for breast cancer cases were available, they would have led to better depiction and interpretation of results. In addition, obesity may lead to other diseases or an outcome of some health issues; thus, it is always challenging to explain direct mechanisms involved in its association.

## 5. Conclusions

In conclusion, increased age and obesity contribute to substantial breast cancer MRs and DALYs in China and around the globe. These results support the promotion of exercise and proper diet in Chinese women in order to provide help in disease prevention and management.

## Figures and Tables

**Figure 1 ijerph-17-01367-f001:**
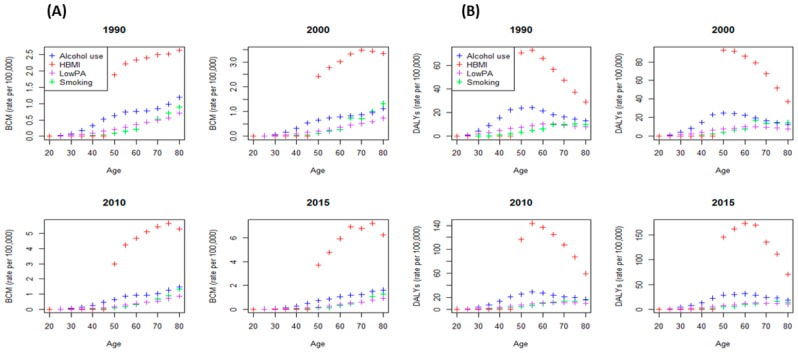
Trends in breast cancer mortality (BCM) and disability-adjusted life years (DALYs) across ages within different risk factors (high body mass index (HBMI), alcohol use, low physical activity (PA), and smoking) in years 1990, 2000, 2010, and 2015. (**A**) The left panel indicate the BCM plots, and (**B**) the right panel indicate the breast cancer DALYs plots.

**Figure 2 ijerph-17-01367-f002:**
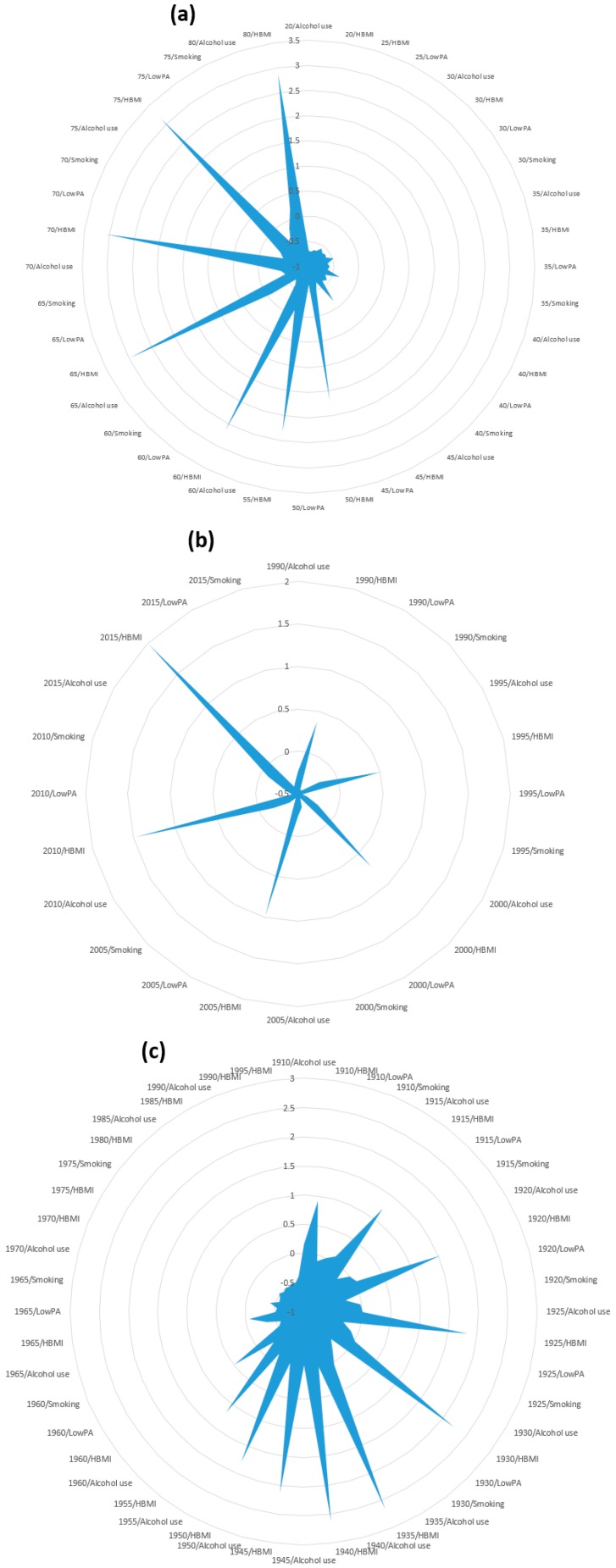
Random effects of (**a**) age, (**b**) period, and (**c**) cohort interaction with different risk factors on mortality rates (MRs) using hierarchical age–period–cohort (APC) model.

**Figure 3 ijerph-17-01367-f003:**
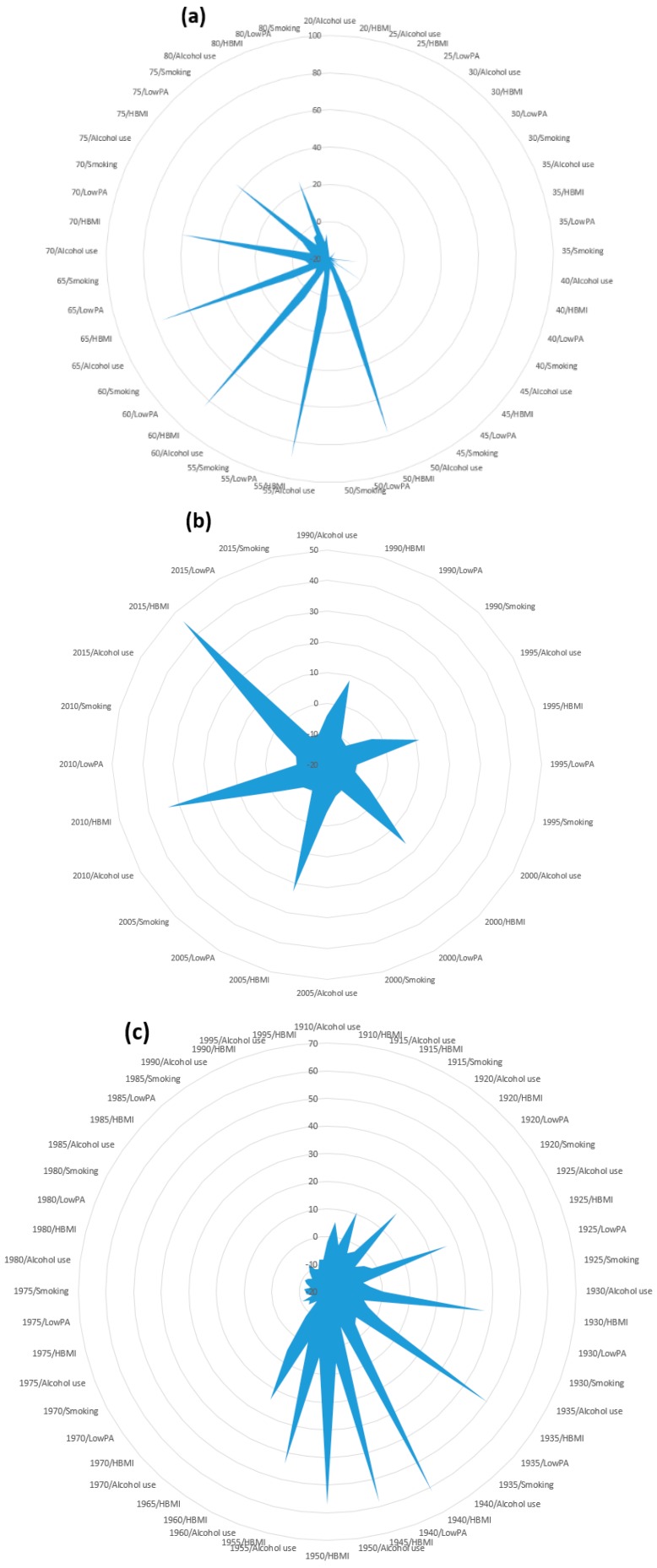
Random effects of (**a**) age, (**b**) period, and (**c**) cohort interaction with different risk factors on DALYs using hierarchical APC model.

**Figure 4 ijerph-17-01367-f004:**
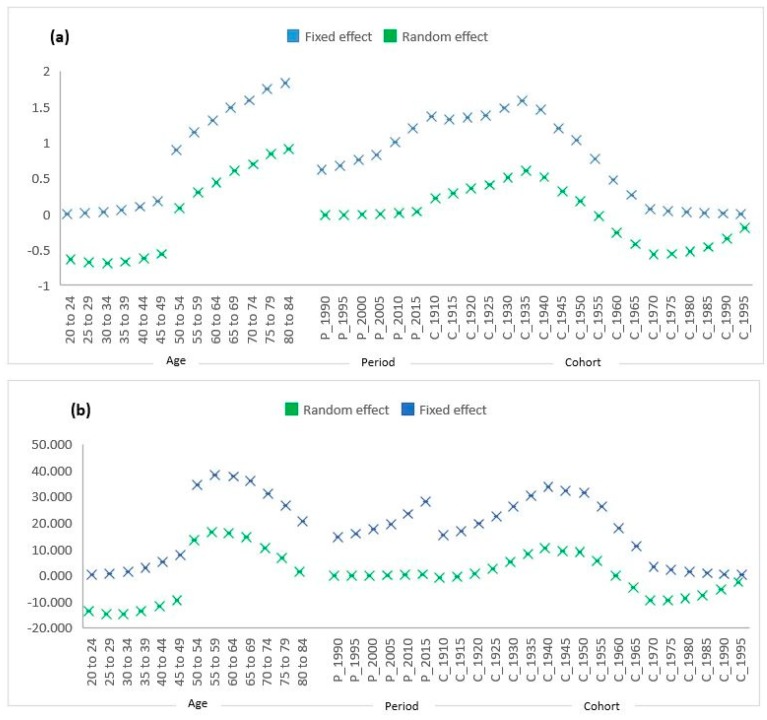
Fixed and random effects of age, period, and cohort from hierarchical (APC) model of (**a**) MRs and (**b**) DALYs (per 100,000) from breast cancer.

**Table 1 ijerph-17-01367-t001:** Generalized linear model (GLM) and mixed-effect model estimates (*p*-value) predicting breast cancer mortality (BCM) among Chinese women. AIC—Akaike information criterion; BIC—Bayesian information criterion.

**Predictors**		**GLM**		**Mixed Effect Models**		
		**Model** **Estimate**		**Model 1** **Estimate**	**Model 2** **Estimate**	**Model 3** **Estimate**
(Intercept)		−0.164 (0.591)		**−1.733 (0.000)**	**−1.128 (0.006)**	−0.164 (0.662)
Age		**0.011 (0.046)**		**0.042 (0.000)**	**0.030 (0.001)**	**0.010 (0.0450)**
Period (Reference 1990)						
1995		0.074 (0.840)		0.057 (0.576)	0.074 (0.797)	0.074 (0.798)
2000		0.209 (0.568)		0.135 (0.186)	0.209 (0.468)	0.209 (0.471)
2005		0.344 (0.349)		**0.215 (0.036)**	0.344(0.233)	0.344 (0.236)
2010		0.663 (0.072)		**0.392 (0.000)**	**0.662 (0.022)**	**0.663 (0.023)**
2015		**0.950 (0.010)**		**0.587 (0.000)**	**0.950 (0.001)**	**0.950 (0.001)**
Risk (Reference: alcohol use)						
HBMI		**2.155 (0.000)**		**1.438 (0.000)**	**1.437 (0.000)**	**2.154 (0.000)**
Low PA		0.096 (0.744)		−0.394 (0.191)	−0.394 (0.195)	−0.096 (0.848)
Smoking		−0.354 (0.277)		−0.435 (0.160)	−0.435 (0.164)	−0.354 (0.526)
Age × 1995		0.003 (0.705)			0.002 (0.630)	0.003 (0.632)
Age × 2000		0.007 (0.321)			0.006 (0.207)	0.007 (0.210)
Age × 2005		0.011 (0.108)			**0.010 (0.041)**	**0.011 (0.042)**
Age × 2010		**0.020 (0.003)**			**0.020 (0.000)**	**0.020 (0.000)**
Age × 2015		**0.030 (0.000)**			**0.029 (0.000)**	**0.030 (0.000)**
Age × HBMI		**0.072 (0.000)**				**0.072 (0.000)**
Age × Low PA		−0.008 (0.120)				−0.008 (0.365)
Age × Smoking		0.000 (0.962)				0.000 (0.978)
**Model Selection Parameter Estimates among GLM and Mixed-Effect Models**						
**Model**	**AIC**	**BIC**	**Log Likelihood**	**Test**	**L Ratio**	***p*-Value**
GLM	545.1321	615.1202	−253.566			
Model 1	573.7778	621.6644	−273.889	GLM vs. 1	40.64567	<0.0001
Model 2	538.8635	605.0037	−251.432	1 vs. 2	44.86496	<0.0001
**Model 3**	**485.8985**	**563.2537**	**−** **221.949**	**2 vs. 3**	**59.0393**	**<0.0001**

Notes: Data from Global Burden of Disease (GBD) study in 2017; bold values denote statistical significance at *p* < 0.05.

**Table 2 ijerph-17-01367-t002:** GLM and mixed-effect model estimates (*p*-value) predicting disability-adjusted life years (DALYs) among Chinese women.

**Predictors**			**GLM**	**Mixed Effect Models**		
			**Model** **Estimate**	**Model 1** **Estimate**	**Model 2** **Estimate**	**Model 3** **Estimate**
(Intercept)			5.143 (0.635)	**−24.939 (0.020)**	−13.866 (0.232)	5.143 (0.735)
Age			**0.103 (0.040)**	**0.698 (0.002)**	**0.483 (0.029)**	**0.104 (0.042)**
Period (Reference: 1990)						
1995			0.836 (0.949)	1.381 (0.565)	0.836 (0.905)	0.836 (0.905)
2000			3.020 (0.817)	3.098 (0.197)	3.019 (0.667)	3.019 (0.669)
2005			5.591 (0.669)	**4.953 (0.039)**	5.590(0.426)	5.591 (0.429)
2010			**10.290 (0.043)**	**8.934 (0.000)**	**10.290 (0.0144)**	**10.290 (0.015)**
2015			**14.734(0.026)**	**13.600 (0.000)**	**14.734 (0.036)**	**14.734 (0.037)**
Risk (Reference: alcohol use)						
HBMI			**37.666 (0.000)**	**31.728 (0.000)**	**31.728 (0.000)**	**37.666 (0.006)**
Low PA			−2.368 (0.822)	−10.199 (0.247)	−10.198 (0.251)	−2.440 (0.909)
Smoking			−10.349 (0.374)	−11.891 (0.190)	−11.891 (0.194)	−11.162 (0.641)
Age × 1995			0.043 (0.858)		0.042 (0.738)	0.043 (0.739)
Age × 2000			0.118 (0.621)		0.118 (0.357)	0.118 (0.360)
Age × 2005			0.204 (0.394)		0.203 (0.113)	0.204 (0.115)
Age × 2010			0.372 (0.121)		**0.371 (0.004)**	**0.372 (0.004)**
Age × 2015			**0.548 (0.023)**		**0.547 (0.000)**	**0.548 (0.000)**
Age × HBMI			**1.388 (0.000)**			**1.388 (0.000)**
Age × Low PA			−0.131 (0.497)			−0.129 (0.741)
Age × Smoking			0.007 (0.975)			0.0193 (0.963)
**Models Selection Parameters Estimates among GLM and Mixed Effect Models**						
**Model**	**AIC**	**BIC**	**Log** **Likelihood**	**Test**	**L. Ratio**	***p*-Value**
GLM	2646.76	2716.74	−1304.38			
Model 1	2451.29	2499.18	−1212.64	GLM vs. 1	183.46	<0.0001
Model 2	2434.55	2500.86	−1199.27	1 vs. 2	26.73	<0.0001
**Model 3**	**2422.40**	**2499.76**	**−** **1190.20**	**2 vs. 3**	**18.14**	**<0.0004**

Notes: Data from GBD study in 2017; bold values denote statistical significance at *p* < 0.05.

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
