# Peer review of "A Hierarchical Age–Period–Cohort Analysis of Breast Cancer Mortality and Disability Adjusted Life Years (1990–2015) Attributable to Modified Risk Factors among Chinese Women"

_ijerph, 2020, doi:10.3390/ijerph17041367_

Round 1
Reviewer 1 Report
Line 44-45: What does “Older breast cancer cases…” means? The word “older” are you referring to the age of the patient or the year of the cancer case? Perhaps the problem is the word “cases”, may be it should be “patients”.
Line 57: What is “lac”? Please don’t use abbreviation without stating the full version of the word.
Line 78-80: This sentence reads weird. It needs to be rephrased.
Line 85-94: The authors did not mention anything about which country’s GBD data was used in this study. Please clearly state which data set was used for this study, country, period, sex, age…etc.
Line 148: “…Chinese adults…”. Are you referring to both male and female? Or are you referring to Chinese women?
Figure 1: Why each item on the X-axis has multiple Y-axis values? E.g. Why the age 50 group has multiple death rates?
Figure 1: Why there are no HBMI in age groups below 50? It is impossible to have 0 HBMI within these age groups.
Figure 1: What is the number on the X-axis Cohort referring to? What does cohort mean here? The same problem in multiples lines mentioning “cohort”, e.g. line 217-218.
Line 149: Is the “MRs” here refereeing to the DRs? And the Y-axis for Figure 1 are labeled as Death-Rate. Please keep the terms consistent throughout the manuscript.
Figure 2 and 3: Labels on the Y-axis are extremely difficult to see.
Reviewer 2 Report
IJERPH-724029 presents results for DALY breast cancer. While some parts of this manuscript were interesting, other areas could be improved. I hope the authors consider my feedback for enhancing their manuscript.
MAJOR COMMENTS
Data Source: In the Introduction, obesity was discussed as a risk factor at 30 kg/m2; however, the overweight threshold (25kg/m2) is being used here instead. Please revise to 30kg/m2 here in the data source or revise to the impact of “overweight” in the Introduction and other relevant areas of the manuscript. Methods: More details about how death and age at death was measured is needed in the text. Breast cancer diagnosis details area also welcomed. I know a citation is provided, but details for these variables should be clear because they are the primary variables of interest for this study. Statistical Analysis: A more detailed outline about how DALY was calculated is needed in this section. Moreover, was a prevalence- or incidence-based calculation used? Results: The findings for DALYs would benefit from further evaluating the role of YLL and YLD. While DALYs are important, they don’t show the level of specificity regarding years of life lost or years lived with the disease. Consider including these results in relevant areas where DALY is presented.MINOR COMMENTS
Lines 37-38: Consider revising to, “Considering lifestyle factors such as obesity, extravagant…”. Line 40: Should be, “researchers”. Be sure to double-check for typos and minor grammatical edits throughout. Likewise, I am unsure what “load of death rates” is meaning? Line 92: “DR” has not yet been defined. Figures 2A and C; Figure 3A and C are not readable. Figure 4: The effect trajectories should not run into the age and cohort keys. A better presentation is needed here. Moreover, is there a reason that explains the random effects here? Make any changes to the abstract that align with those made in the text.Author Response
Please see the attachment.

Round 2
Reviewer 2 Report
The authors have done a nice job addressing my previous concerns; however, the Figures could still be improved. For example, text overlaps in Figures 2 and 3 (parts a and c), and the findings in Figure 4 still intersect with axis labels (random effect). Consider a better presentation for these figures to further improve your manuscript.
